# Accurate Flow Characterization of A6082 for Precision Simulation of a Hot Metal Forming Process

**DOI:** 10.3390/ma15238656

**Published:** 2022-12-05

**Authors:** Jeong-Hwi Park, Su-Min Ji, Jeong-Muk Choi, Man-Soo Joun

**Affiliations:** 1Material Manufacturing Engineering Team, Hyundai Wia, Uiwang 16082, Republic of Korea; 2Graduate School of Mechanical and Aerospace Engineering, Gyeongsang National University, Jinju 52828, Republic of Korea; 3Jinhap Co., Ltd., Daejeon 34302, Republic of Korea; 4Engineering Research Institute, School of Mechanical and Aerospace Engineering, Gyeongsang National University, Jinju 52828, Republic of Korea

**Keywords:** aluminum alloy, flow characterization, friction compensation, temperature compensation

## Abstract

The flow behaviors of metallic materials are sensitive to state variables, including strain, strain rate, and temperature. In particular, the temperature effect on the flow behavior is of great importance. The flow information is usually obtained at the sample strain rates and temperatures from the hot cylinder compression test. However, this test is inevitably exposed to undesirable effects of friction and temperature on flow characterization. This study reveals their impact on the flow curve of an A6082 alloy. The unique features of its flow behavior and the inaccuracy of as-received, primitive flow information are emphasized. Using a systematic way of correcting the friction and temperature effects, the flow curves with high accuracy in terms of the compression load–stroke curve obtained from the test are calculated. It was revealed that the both the friction and temperature compensation of the primitive flow curves bring a minor change in the flow curves of the A6082 alloy, which is quite different from other commercial light metals. This phenomenon caused by the unique features of the flow behavior of the A6082 or other aluminum alloys will be critical to solving various process and quality matters confronted by the engineers in the hot metal forming industry.

## 1. Introduction

Aluminum alloys are one of the attractive metallic materials, satisfying the demands of the aircraft and automobile industries, particularly for weight reduction of the vehicle. They have good mechanical and metallurgical features, including low density, weldability and machinability, and recycling capability [1,2,3]. Aluminum alloy parts with strength and integrity have traditionally been forged for structural members. Notably, the aluminum billets, fabricated mostly by extrusion of cast materials with limited area reduction, have no or poor metal flow lines compared to steel bars, which have distinct metal flow lines. Forging aluminum alloys contributes to forming metal flow lines that are friendly to their strength [4].

Aluminum alloys exhibit unique flow behaviors, which affect much on metal flow lines, surface quality, etc. For example, they do not experience distinct strain softening and significant change in their flow stresses owing to increased temperature [5,6,7] compared to the other commercial metals [8]. Understanding their flow behaviors and mathematical description with high accuracy is thus essential [9]. During the experimental acquisition of flow curves, using a specific material testing method, the effects of friction and temperature are pretty influential. They should be corrected for the flow curves with acceptable accuracy [10]. In addition, mathematically describing the flow curves with accuracy, flexibility, and practicability is also important because the flow behavior exhibits high non-linearity with strain, strain rate, temperature, etc.

It has been known that aluminum alloys are susceptible to their chemical compositions and the working conditions of state variables. The flow behaviors of aluminum alloys are thus too complicated to simply formulate in a mathematical form. Even though many kinds of research on their flow characterization have been metallurgically or phenomenologically conducted for several decades, based on specific mathematical model equations [9], they do not meet the need of the industries because of poor accuracy.

A few general flow stress models targeted to cover the whole range of the state variables of interest have been presented, including the Johnson–Cook model [11], Khan et al. model [12], Zhang et al. model [13], Samantaray et al. model [14], Gao and Zhang model [15], Rusinek–Klepaczko model [16], etc. They showed suitable fittings for a wide range of strain rates, for example, from 0.001 s^−1^ to 3700 s^−1^ [17]. However, they may not be appropriate to model the specific flow behaviors of aluminum alloys to be hot formed at elevated temperatures.

On the contrary, the constitutive models considering dynamics recrystallization can be applied to characterize the flow behaviors of aluminum alloys to be hot formed. The Voce model [18] represents them. However, its formulation is too complicated to identify its flow constants practically. Ebrahimi et al.’s model [19] inherently describes recrystallization well. However, the flow constants in this model should be graphically determined. To cope with this matter, Razali et al. [20] suggested a systematic way of calculating the flow constants employed to define the flow parameters as the functions of the state variables.

Furthermore, some phenomenological descriptions using mathematical models can be employed to fit the flow behaviors of aluminum alloys to be plastically formed, including the Arrhenius model and its improvements [7,20], Fields–Backofen model [21], Hensel–Spittel model [22], and *C-m* model [23]. The hyperbolic sine Arrhenius (or Garofalo-Arrhenius) model [24] is simple but does not involve the strain term as the primary parameter. Thus, the flow parameter should be formulated as a function of strain to reflect the strain effect on the flow behaviors. However, the primary material parameters should be prepared as functions of high-order polynomials of strain to achieve acceptable accuracy [25,26].

Sutton and Luo [25] proposed an extended Hollomon model, i.e., the Fields–Backofen model, to obtain an acceptable result. The Hensel–Spittel model is simple but inaccurate for complicated aluminum alloys because of the limited number of flow constants. Joun et al. [23] recently presented a generalized *C-m* model with the piecewise description scheme of many flow constants. Recently, various artificial-neural-network models have been studied to characterize the hot deformation behaviors of metallic materials [27,28,29,30], and they are deemed promising methods for solving the hottest issue in this field.

It should be noted that the flow stress at an instant should be obtained under the assumption of isothermal and homogeneous cylinder compression. Acquisition of accurate flow information from the material testing should thus be preceded by the correction of the friction and temperature effects on the flow test. Luan et al. [10] presented an experimental rule-based approach to reflecting the friction and temperature effects on flow curves in characterizing the flow behavior of a magnesium alloy, using the cylinder compression test. They showed that friction significantly influences the flow curves more than temperature. On the contrary, Joun et al. [31] showed that the AISI 1025 in the cold forging temperature range, Ti-6Al-4V titanium alloy in the warming forging temperature range [8], and magnesium alloys in the hot forging temperature range [32] have no remarkable effects of friction on flow characterization when using the cylinder compression test because the temperature effect overwhelms the flow behavior.

On the contrary, the detailed effects of friction and temperature on the flow characterization of aluminum alloys, using the cylinder compression test, were not revealed. Here, the generalized piecewise *C-m* model [23] is applied to characterize an A6082 alloy with high accuracy, with an emphasis on friction and temperature compensation. After verifying the validity of the compensated flow models comparing the experimental and predicted compression tests, the effects of the flow behavior of the A6082 alloy on the hot metal forming process are qualitatively discussed. For all the finite element analyses of the cylinder compression tests, an implicit thermoviscoplastic FE package AFDEX (Altair APA) [33] was used, based on the tetrahedral MINI-element scheme [34,35] with automatic adaptive remeshing [36]. The die or tool deformation and elastic deformation of material were assumed to be negligible. The material–tool interface was formulated by the law of Coulomb friction, with a friction coefficient of 0.35.

## 2. Experiments

The material studied is an A6082 alloy (chemical composition: Mn (0.4–1.0%), Fe (0.0–0.5%), Mg (0.6–1.2%), Si (0.7–1.3%), Cu (0.0–01%), Zn (0.0–0.2%), and Al (Bal.)). Its melting temperature is 555 °C. It is manufactured by a continuous casting process, with a peeling process to improve its surface.

Solid cylinder compression tests were conducted by using a Gleeble 3500 system. The diameter and height of the specimen are 10 and 15 mm, respectively. A specimen for each test case was tested since the test system was adequately and strictly managed and well-controlled [31]. The sample test temperatures (°C) and strain rates (s^−1^) were 350, 400, 450, 500, and 550 and 0.1, 1.0, 5.0, 10.0, and 20.0, respectively. A tantalum foil lubricated the interface between the specimen and the compression tool. A thermocouple was welded to the surface at the mid-plane to monitor the specimen temperature. However, the measured temperatures were used only to validate the calculated flow model. Some selected specimens at the stroke of around 6.3 mm are shown in Figure 1, and their measurements of the height (*h*) and minimum (*d_n_*) and maximum (*d_x_*) diameters at the final stroke are listed in Table 1.

The compression test at 450 °C and a strain rate of 10 s^−1^ was selected for checking the validity of the calculated flow curves in the next section. Note that the measured maximum diameter of the compressed specimen was 13.7 mm at the final stroke of 6.3 mm.

Solid lines in Figure 2 describe the experimental compression load–stroke curves (experimental CLSCs). The dashed lines with square or circle marks in Figure 2 represent the CLSCs obtained by isothermal and non-isothermal flow analyses of the cylinder compression test, respectively, using the primitive flow curves directly calculated from the experimental CLSCs (see the RFCs–BFTC in Figure 3). Note that the oscillation of the compression load of the specimen with the initial temperature (550 °C) near the melting temperature can be observed in Figure 2 as the strain rate and strain increases. The maximum measured temperature at the barreled surface reached 557 °C, implying that internal local melting might occur during the compression test at the strain rate of 20 s^−1^.

Solid lines in Figure 3 exhibit the approximate primitive flow curves for the sample temperatures and strain rates, termed the reference flow curves before friction compensation and temperature compensation (RFCs–BFTCs), which were calculated under the assumption of isothermal and homogeneous cylinder compression, viz without any compensation for friction and temperature effects on flow behavior.

The flow curves in Figure 3 are characterized by their negligible slopes after peak strains of less than 0.1, regardless of sample strain rates and temperatures. Note that the drop rate of flow stress to temperature is low. The flow stress drop rate of the A6082 alloy, owing to the unit temperature increase at 450 °C temperature, 0.3 strain, and 10 s^−1^ strain rate, is −0.09 (MPa/°C). These behaviors are quite different from those of other light metals. For example, the flow stresses of the Ti-6Al-4V alloy (>600 °C) [8] and AZ80A alloy (>250 °C) [32] start to drop from the peak strains distinctly. In addition, the flow stress drop rates of the Ti-6Al-4V alloy (at 700 °C temperature, 0.3 strain, and 10 s^−1^ strain rate) and AZ80A alloy (at 250 °C temperature, 0.3 strain, and 10 s^−1^ strain rate) are −1.10 and −0.78 (MPa/°C), respectively.

The fitted RFCs–BFTCs (dashed curves) of the experimental RFCs–BFTCs in Figure 3 were obtained by using the following generalized piecewise *C-m* flow model [23]:(1)$$\sigma =\left(C_{1}+C_{2}\ln\;\tilde{\dot{\varepsilon }}\right){\dot{\varepsilon }}^{m_{1}+m_{2}\ln\;\tilde{\dot{\varepsilon }}}$$
where *C*_1_, *C*_2_, *m*_1_, and *m*_2_ are all flow constants defined at the fixed sample strains and temperatures, and $\tilde{\dot{\varepsilon }}$ is the maximum value between 0.1 and the strain rate, denoted by $\dot{\varepsilon }$. These flow constants were numerically calculated by minimizing the errors between the fitted and experimental RFCs–BFTCs; they are listed in Table 2. Some details about calculating the flow constants can be seen in Ref. [23].

## 3. Friction and Temperature Compensation

The average absolute relative error (AARE) and root mean square error (RMSE) of the fitted RFCs–BFTCs shown in Figure 3 over the strain range from 0.05 to 0.45 are 2.0% and 1.7 MPa, respectively, implying that the fitted flow curves are pretty accurate. However, it should be emphasized that the fitted flow curves in Figure 3 are primitive and need a kind of state variable compensation because they were calculated under the assumption of isothermal and homogeneous cylinder compression. The temperatures at any material points of the specimen during experiments are unknown, although those at the specific material points can be measured. As the frictional effects during the cylinder compression test cause non-uniformities of the strain, strain rate, and temperature, the plastic deformation of the materials locally heats themselves depending on their flow behavior, thermal capacity, and thermal conductivity. Notably, the temperature assigned on the flow curve in Figure 3 means the initial sample temperature of the specimen, which is fixed. Friction and temperature compensations [8,10,32] should thus be made to reveal the actual flow behavior of the material.

The dashed curves in Figure 2, the predicted CLSCs, were obtained using a rigid-thermoviscoplastic finite element method [33,37] with the fitted RFCs–BFTCs and the thermal material properties and process conditions found from the reference [33,38,39]. Notably, the difference between the experimental and predicted CLSCs is relatively tiny compared to the Ti-6Al-4V and AZ80A alloys [8,32], even though a distinct difference exists.

As can be seen from Table 3, the average values of mean and maximum errors over the stroke range from 1.0 to 5.5 mm are 1.80% and 3.43%, respectively. They are pretty small. This phenomenon is quite exceptional for the A6082 alloy. To reveal its unique features, the validity of the assumption of homogeneity in the cylinder compression test to obtain the RFCs–BFTCs was first examined.

The frictional coefficient was presumed at 0.35 by comparing the barreled specimens’ measured diameters with their FE predictions by the fitted RFCs–BFTCs and measured velocities. Notably, the presumed frictional coefficient is great because the barreling might be affected much by both non-uniformity of the initial temperature and the viscous heating, which exaggerates the frictional phenomena.

Figure 4 compares the predicted CLSCs of the two frictional coefficients of 0.0 and 0.35 at the strain rate of 0.1 s^−1^. The percent difference in the predicted CLSC ranges from 2.6% to 4.5% for all the test cases, revealing that the friction effect on flow behavior may be meaningful, even though it is negligible when the temperature effect is great [8,32].

From Figure 4, the linear change of the percent difference (%) of CLSC with the stroke, that is, the friction-compensation coefficient function denoted as *D*(*T,s*), was fitted as a function of strain rate and temperature as follows:(2)$$D\left(T,s\right)=100-\left(0.000244\;T+0.36\right)\left(1+0.052\ln\left(\dot{\varepsilon }\right)+0.12\right)\;s\;(\%)$$
where $\dot{\varepsilon }$, *T*, and *s* are the strain rate, temperature, and stroke, respectively.

The experimental CLSCs were modified by multiplying them by *D*(*T,s*)/100 to obtain the friction-compensated reference flow curves (RFCs–AFCs). Figure 5 compares the primitive flow curves and the RFCs–AFCs, indicating that the friction compensation decreases the flow stress’s magnitude. Notably, the error between experimental flow curves and RFCs–AFCs increases as the strain rate increase, owing to a relatively considerable oscillation of experimental flow curves caused by the increased temperature non-uniformity of the specimen.

The predicted CLSCs by the RFCs–AFCs were compared with the experimental CLSCs in Figure 6. The error analyses summarized in Table 4 indicate that the error inevitably increases after the friction compensation.

Now, the temperature effect is considered. The temperature-compensation algorithm developed previously [32] can be stated as follows: when the sample temperatures and strains are selected in the matrix form, the increased temperature at the *i*-th sample strain ($\varepsilon_{i}$) and *j*-th sample temperature ($T_{j}$), denoted as $T_{j}^{i}$, is approximated by
(3)$$T_{j}^{i}=T_{j}+\xi \frac{\bar{\sigma }\varepsilon_{i}}{\bar{\rho }\bar{c}}$$
where $\bar{\rho }\bar{c}$, $\bar{\sigma }$, and ξ are the average volumetric thermal capacity, average flow stress, and instantaneous temperature correction parameter (TCP), respectively. The TCP depends on the unit system, the energy dissipation ratio to the stress power, strain rate, and thermal conductivity. In general, the smaller the thermal conductivity, or the greater the strain rate, the nearer to unity the TCP is. The titanium alloy has a relatively great TCP, whereas the magnesium alloy stays in its opposite position [8,32]. The TCP values at the strain rates of 0.1, 1, 5, 10, and 20 s^−1^ were empirically assumed at 0.0, 0.7, 0.8, 0.9, and 0.9, respectively.

The above algorithm was applied to improve the RFCs–AFCs in Figure 5 and obtain the friction- and temperature-compensated reference flow curves (RFCs–AFTCs), of which flow constants were optimally fitted and listed in Table 5. The RFCs–AFTCs are given in Figure 7 together with the RFCs–BFTCs, notably showing that they are almost the same.

Figure 8 shows the material’s predicted temperature and effective strain at the conditions of 450 °C temperature and 10 s^−1^ strain rate, using the RFCs–AFTCs. It is noted that the experimental temperature at the measurement point is 461 °C, while its corresponding prediction is 462 °C, implying that the RFCs–AFTCs are quite accurate.

Figure 9 also compares the predicted and experimental deformed shapes of the same cylinder compression with Figure 8, revealing that they are in good agreement with each other. Figure 10 compares the experimental CLSCs with their predictions obtained by using the RFCs–AFTCs for the test at the 450 °C temperature and 10 s^−1^ strain rate. Table 6, analyzing the errors of the predicted CLSCs by RFCs–AFTCs, shows that the friction compensation and temperature compensation reduced the errors compared to the RFCs–AFCs.

## 4. Discussion

A comparison of the RFCs–AFCs summarized in Table 4 with the RFCs–BFTCs in Table 3 shows that only friction compensation deteriorates the accuracy of the modified flow curves. However, the comparison of the RFCs–AFTCs summarized in Table 6 with the RFCs–AFCs in Table 4 shows considerable improvement of flow curves by temperature compensation after friction compensation.

However, it is interesting to note that the improvement of the RFCs–AFTCs from RFCs–BFTCs summarized in Table 3 can be neglected, implying that the increase in flow stress by the temperature compensation (2.22% in the case of frictional coefficients of 0.35, strain of 0.46, strain rate of 1 s^−1^, and temperature of 350 °C) is almost the same as its decrease by the friction compensation (2.49% in the same case with above).

The negligible difference in the percent errors of fitted flow curves between the RFCs–BFTCs (1.99%) and RFCs–AFTCs (2.20%) of the A6082 alloy implies that the friction and temperature effects on its flow curve compensate for each other. Notably, this fact is one of the essential flow characteristics of a few aluminum alloys, including the A6082 alloy.

This phenomenon takes place because the A6082 alloy has a particular characteristic of a relatively weak effect of temperature and strain on the variation of flow stress compared to other commercial metals.

The previous study [32] on an AZ80A magnesium alloy showed quite the opposite phenomenon to the A6082 alloy. For the two materials, the effect of friction compensation was nearly the same. Unlike the A6082 alloy, however, temperature compensation has a significant influence on the corrected flow curves of the AZ80A alloy, as shown in Figure 11. Note that we neglected the friction compensation for the corrected flow curves of the AZ80A alloy in Figure 11 because its effect is negligible.

The features of the A6082 alloy revealed in this study, compared to the AZ80A alloy, which is hardly hot forged, are related to its good forgeability. However, the low strain hardening and little temperature-dependent behavior of the corrected flow curves alert us to the possibility of the local instability of the material [31] during plastic deformation, especially near the die–materials interface.

## 5. Conclusions

Based on the cylinder compression test of the A6082 alloy and experimental flow curves of aluminum alloys found in the literature, the most peculiar flow features of aluminum alloys were discussed, with an emphasis on their negligible slopes of flow curves after peak strains less than 0.1, regardless of sample strain rates and temperatures during the cylinder compression test. It was also emphasized that the flow stress drop rates of the A6082 alloy to temperature relative to its melting temperature are pretty low compared to other commercial light materials, including titanium and magnesium alloys. A precision flow characterization of the A6082 alloy was made to reveal the reason for these observations.

A systematic and practical friction- and temperature-compensation scheme was proposed. It was applied to solve the mismatching problem between the actual cylinder compression test affected by friction and temperature and the ideal mathematical calculation of primitive flow curves from the test under homogeneous and isothermal cylinder compression assumptions.

It was revealed that the RFCs–AFTCs are similar to the primitive flow curves, implying that the decrease in flow stress at a specific condition by the friction compensation is almost the same as the flow stress increase after the temperature compensation. This phenomenon is caused by the fact that the flow behaviors of aluminum alloys are less sensitive to the temperature change during plastic deformation because the friction is inherently less influential on flow characterization (about a 2% difference in the case of the A6082 alloy). Notably, this phenomenon is friendly to the hot forgeability of the A6082 alloy and the like. However, such flow behaviors of the A6082 alloy or other similar aluminum alloys may cause a sort of local plastic deformation instabilities, especially at the die–material interface in hot forming with moderate production speed.

## Figures and Tables

**Figure 1 materials-15-08656-f001:**
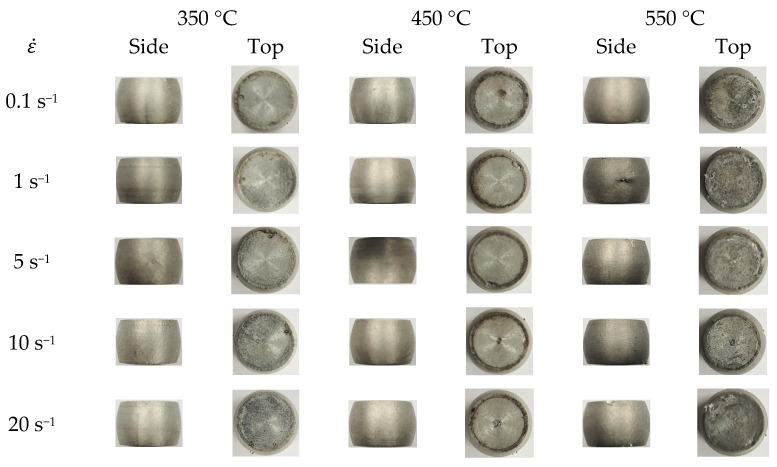
Selected A6082 alloy specimens.

**Figure 2 materials-15-08656-f002:**
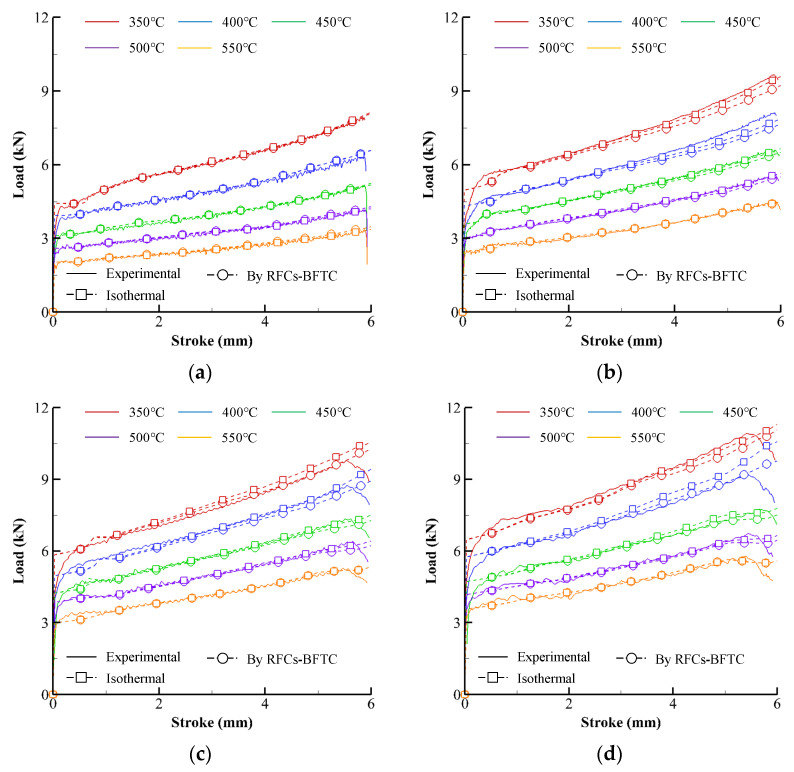
Experimental CLSCs and their predictions by the RFCs–BFTCs in Figure 3: (**a**) 0.1 s^−1^, (**b**) 1 s^−1^, (**c**) 5 s^−1^, (**d**) 10 s^−1^, and (**e**) 20 s^−1^.

**Figure 3 materials-15-08656-f003:**
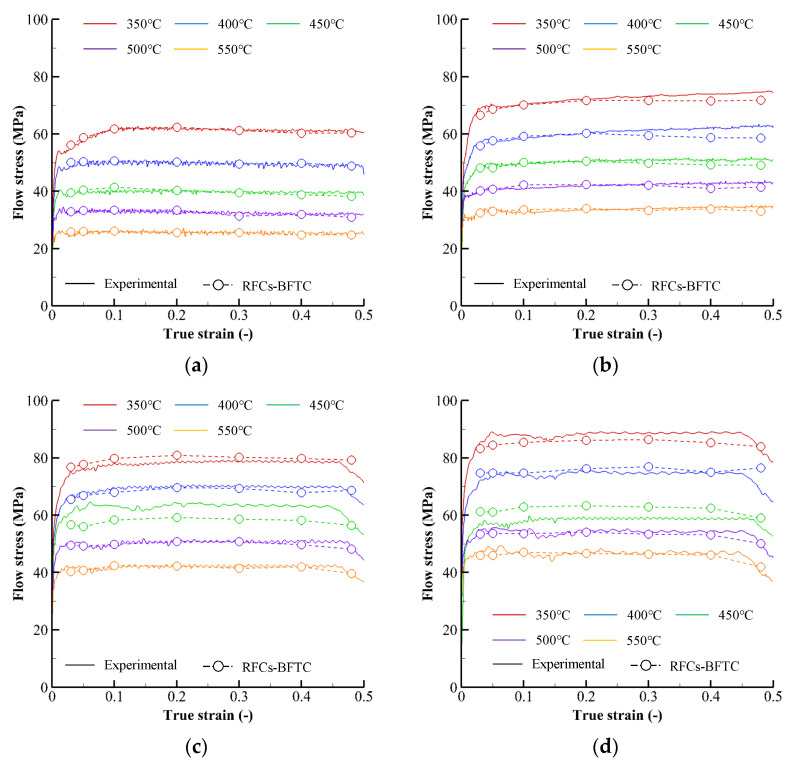
Primitive flow curves, i.e., FRCs–BFTCs: (**a**) 0.1 s^−1^, (**b**) 1 s^−1^, (**c**) 5 s^−1^, (**d**) 10 s^−1^, and (**e**) 20 s^−1^.

**Figure 4 materials-15-08656-f004:**
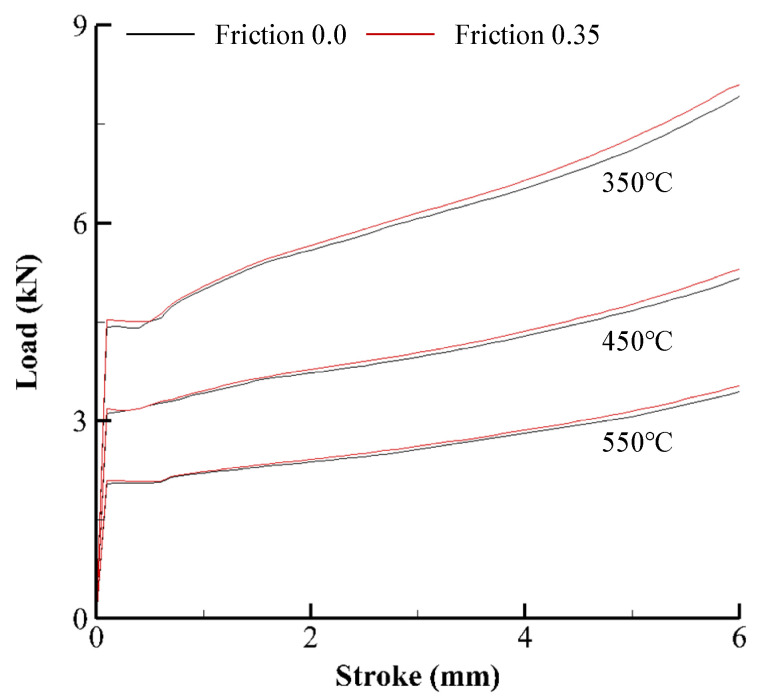
Effect of frictional coefficients on the predicted CLSCs.

**Figure 5 materials-15-08656-f005:**
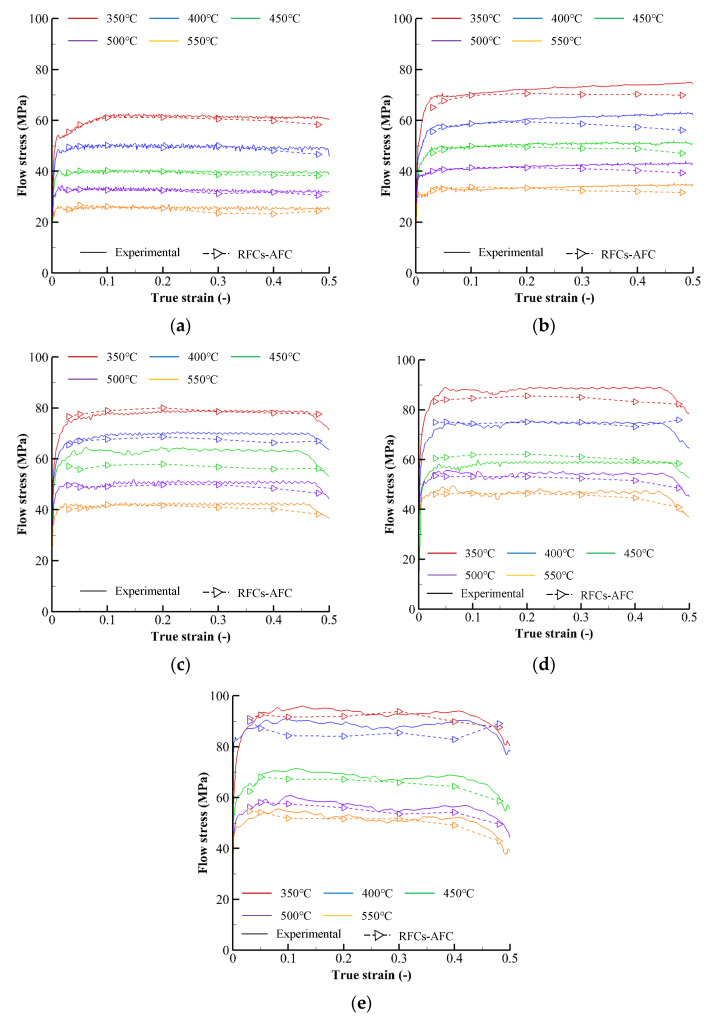
Primitive flow curves and RFCs–AFCs: (**a**) 0.1 s^−1^, (**b**) 1 s^−1^, (**c**) 5 s^−1^, (**d**) 10 s^−1^, and (**e**) 20 s^−1^.

**Figure 6 materials-15-08656-f006:**
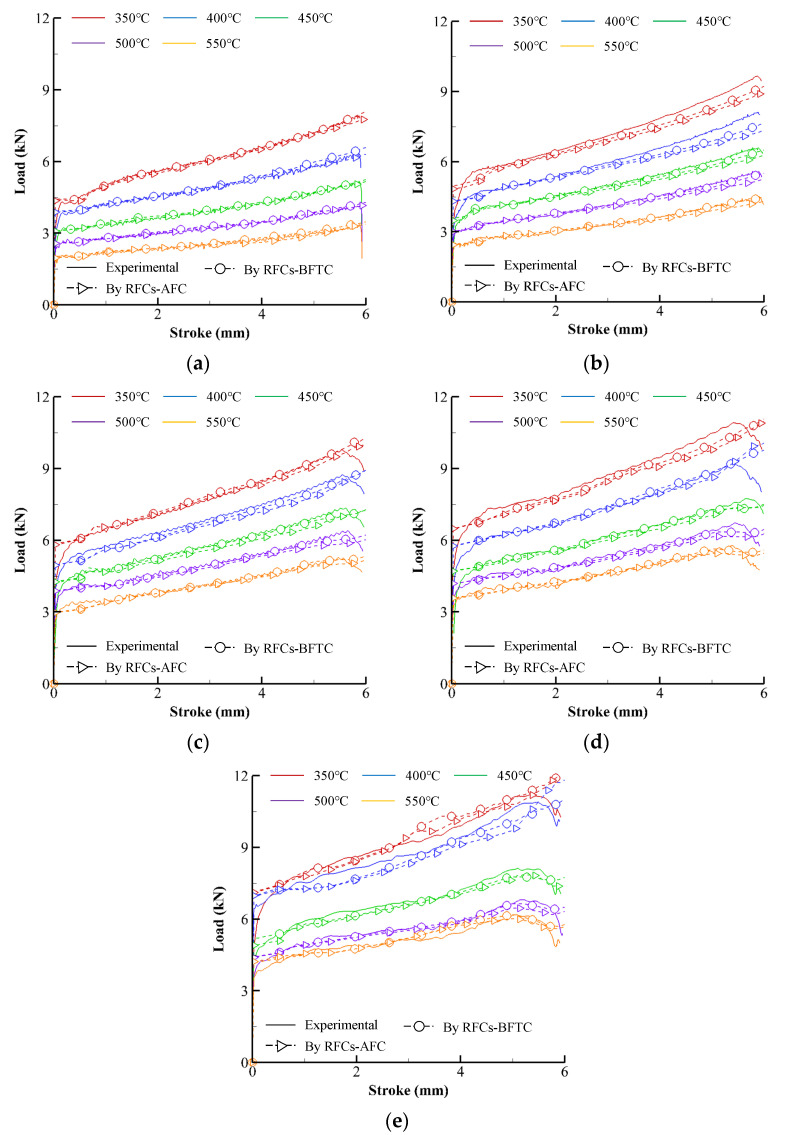
Experimental CLSCs and their predictions by the RFCs–AFCs: (**a**) 0.1 s^−1^, (**b**) 1 s^−1^, (**c**) 5 s^−1^, (**d**) 10 s^−1^, and (**e**) 20 s^−1^.

**Figure 7 materials-15-08656-f007:**
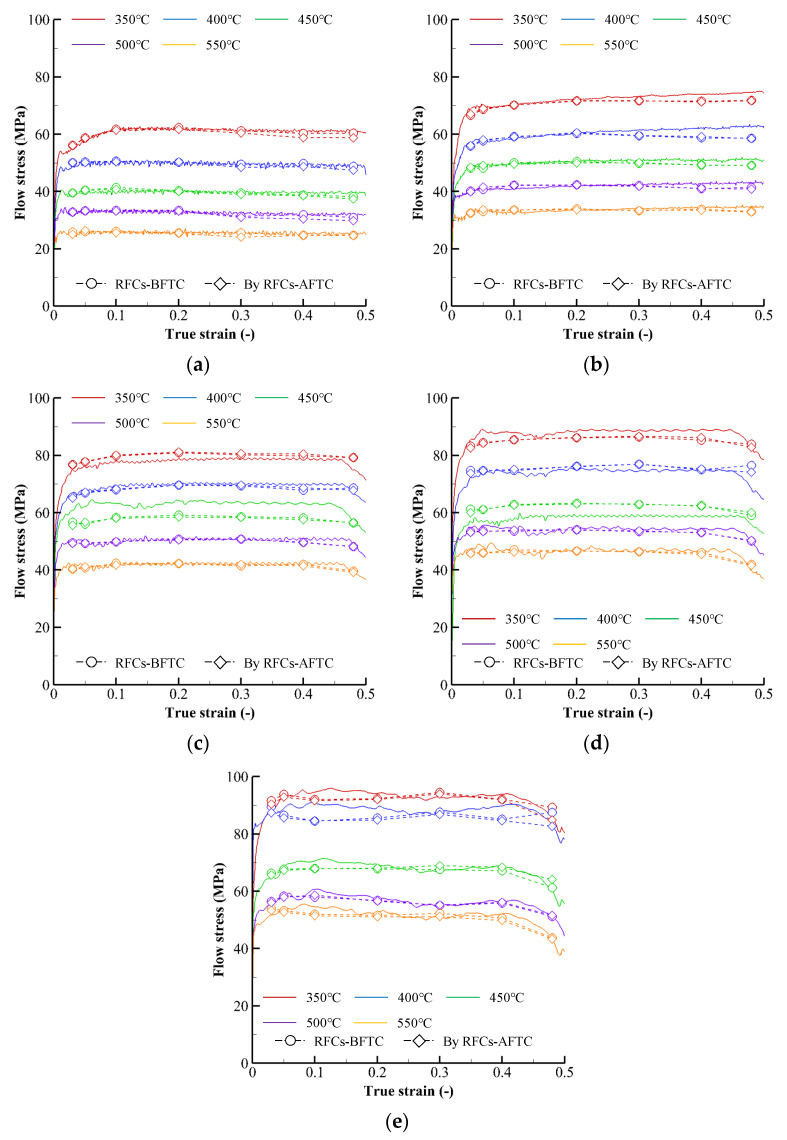
Experimental CLSCs and their predictions by RFCs–AFTCs and RFCs–BFTCs: (**a**) 0.1 s^−1^, (**b**) 1 s^−1^, (**c**) 5 s^−1^, (**d**) 10 s^−1^, and (**e**) 20 s^−1^.

**Figure 8 materials-15-08656-f008:**
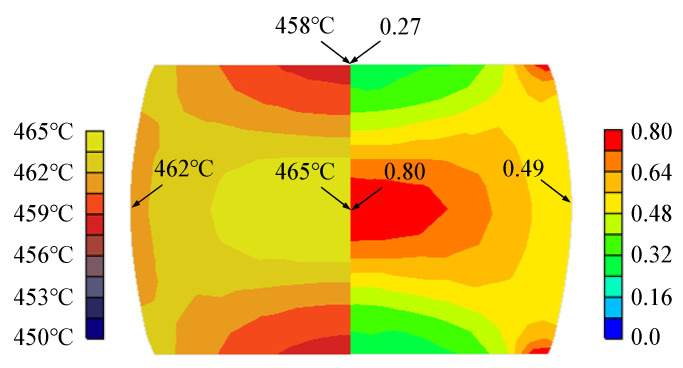
Predicted temperature (**left**) and effective strain (**right**) at the final stroke of cylinder compression test at the 450 °C temperature and 10 s^−1^ strain rate.

**Figure 9 materials-15-08656-f009:**
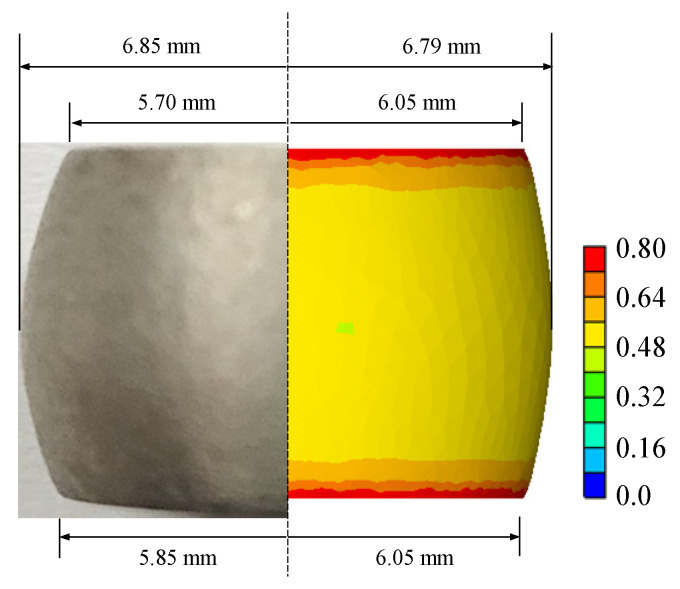
Comparison of the predicted and experimental specimens at the final stroke of cylinder compression test at the 450 °C temperature and 10 s^−1^ strain rate.

**Figure 10 materials-15-08656-f010:**
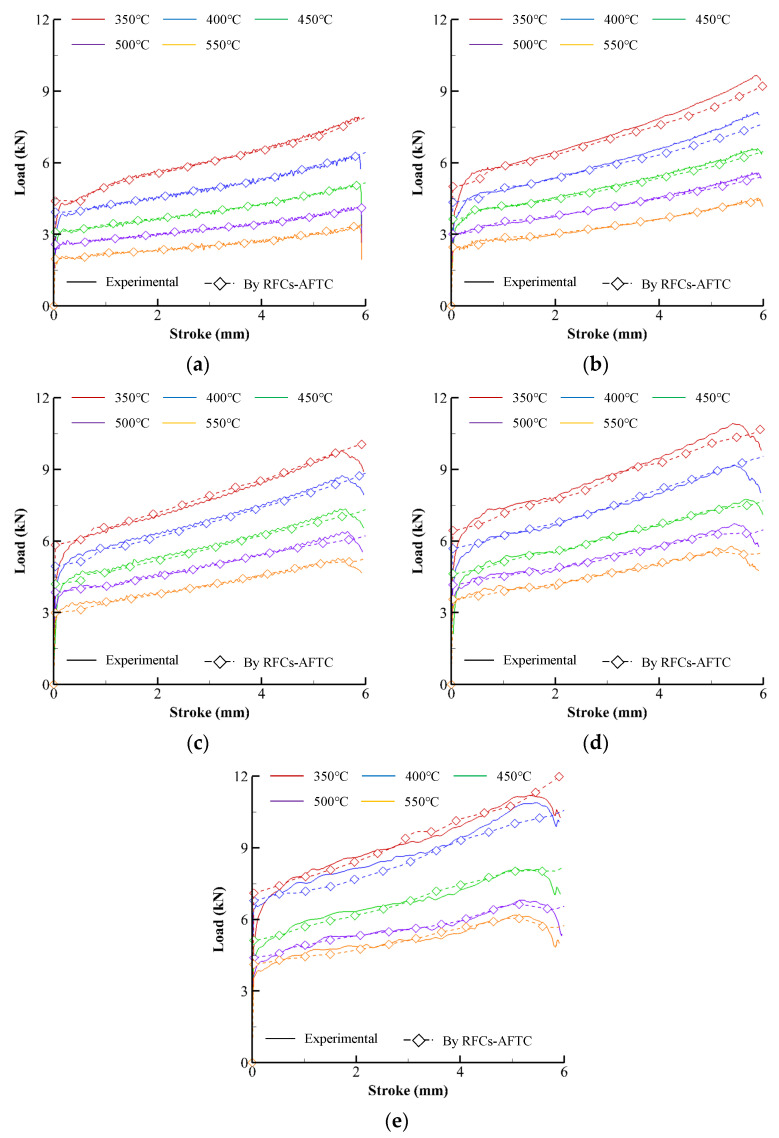
Experimental CLSCs and their predictions by RFCs–AFTCs: (**a**) 0.1 s^−1^, (**b**) 1 s^−1^, (**c**) 5 s^−1^, (**d**) 10 s^−1^, and (**e**) 20 s^−1^.

**Figure 11 materials-15-08656-f011:**
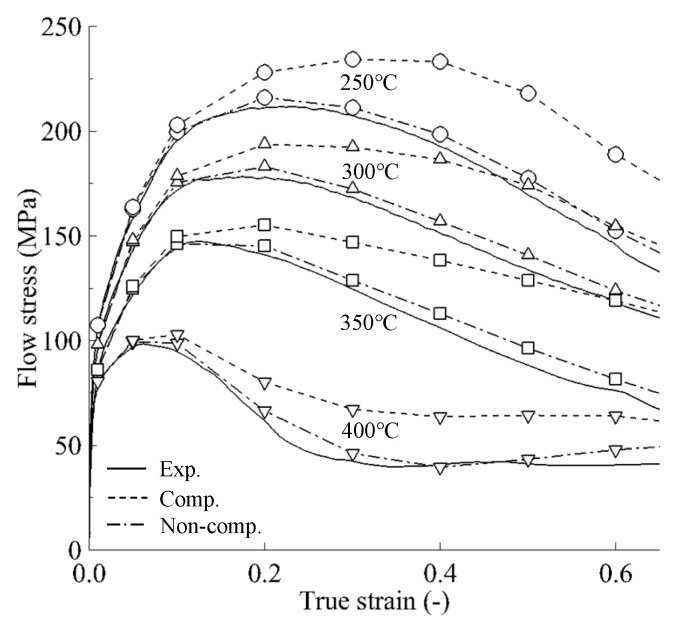
Experimental flow curves and RFCs–ATCs of the AZ80A alloy (strain rate = 10 s^−1^) [32].

**Table 1 materials-15-08656-t001:** Measured height (*h*) and minimum (*d_n_*) and maximum (*d_x_*) diameters of the specimens.

Strain	350 °C			450 °C			550 °C		
Rate (s^−1^)	*h* (mm)	*d_x_* (mm)	*d_n_* (mm)	*h* (mm)	*d_x_* (mm)	*d_n_* (mm)	*h* (mm)	*d_x_* (mm)	*d_n_* (mm)
0.1	9.0	13.5	11.5	9.1	13.7	11.7	8.9	13.7	11.6
1	8.8	13.6	11.7	9.0	13.7	11.7	8.9	13.8	11.6
5	8.8	13.7	11.7	9.0	13.7	11.9	8.9	13.7	11.5
10	8.8	13.7	12.0	8.9	13.7	11.7	8.8	13.8	11.5
20	9.0	13.7	12.0	8.8	13.7	11.7	8.9	13.7	11.7

**Table 2 materials-15-08656-t002:** Flow constants of the fitted RFCs–BFTCs.

T (°C)	*ε*	$C_{1}$	$C_{2}$	$m_{1}$	$m_{2}$
350	0.10	68.7	2.2	0.047	0.006
	0.20	71.4	0.5	0.058	0.007
	0.30	80.6	−13.0	0.108	0.056
	0.40	80.9	−13.5	0.116	0.054
	0.48	81.3	−13.7	0.117	0.051
400	0.10	63.5	−6.2	0.090	0.040
	0.20	65.4	−7.4	0.106	0.041
	0.30	65.4	−8.6	0.110	0.052
	0.40	64.3	−7.9	0.101	0.050
	0.48	65.2	−9.4	0.113	0.058
450	0.10	49.5	0.8	0.078	0.004
	0.20	50.5	0.0	0.098	0.000
	0.30	49.9	0.0	0.101	0.000
	0.40	49.3	0.0	0.103	0.000
	0.48	50.4	−1.7	0.119	−0.005
500	0.10	42.7	−0.7	0.105	0.004
	0.20	39.6	4.0	0.079	−0.016
	0.30	39.6	3.6	0.107	−0.026
	0.40	38.3	4.0	0.085	−0.017
	0.48	41.6	−0.2	0.129	−0.018
550	0.10	31.1	3.6	0.082	−0.003
	0.20	33.2	1.0	0.114	0.001
	0.30	33.1	0.3	0.111	0.011
	0.40	33.8	0.0	0.135	0.000
	0.48	31.8	1.8	0.111	−0.019

**Table 3 materials-15-08656-t003:** Error analysis of the CLSCs predicted by RFCs–BFTCs.

Average Error for All Sample Strain Rates					
Temp.	Mean	Max	Temp.	Mean	Max
350	2.00	4.61	400	2.18	4.67
450	1.56	4.34	500	1.41	4.29
550	1.84	5.32			

**Table 4 materials-15-08656-t004:** Error analysis of the CLSCs predicted by RFCs–AFCs.

Average Error for All Sample Strain Rates					
Temp.	Mean	Max	Temp.	Mean	Max
350	2.52	4.84	400	2.98	6.20
450	2.51	4.88	500	2.25	5.86
550	2.09	5.76			

**Table 5 materials-15-08656-t005:** Flow constants of the fitted RFCs–AFTC.

T (°C)	*ε*	$C_{1}$	$C_{2}$	$m_{1}$	$m_{2}$
350	0.10	67.9	3.3	0.046	0.003
	0.20	70.7	1.3	0.060	0.003
	0.30	81.7	−14.4	0.118	0.061
	0.40	82.3	−16.0	0.132	0.067
	0.48	83.3	−16.7	0.137	0.060
400	0.10	61.6	−3.8	0.082	0.030
	0.20	65.6	−7.4	0.107	0.039
	0.30	65.6	−8.8	0.121	0.049
	0.40	64.9	−8.2	0.114	0.045
	0.48	64.8	−9.0	0.124	0.046
450	0.10	49.2	0.5	0.083	0.005
	0.20	50.4	−0.8	0.099	0.006
	0.30	55.1	−7.2	0.142	0.034
	0.40	53.8	−6.5	0.137	0.031
	0.48	54.2	−7.4	0.152	0.027
500	0.10	42.5	−0.5	0.107	0.004
	0.20	39.4	4.0	0.080	−0.016
	0.30	39.3	3.7	0.110	−0.026
	0.40	41.3	−0.2	0.134	−0.009
	0.48	40.8	0.0	0.138	−0.020
550	0.10	32.9	1.0	0.108	0.004
	0.20	31.7	2.8	0.100	−0.007
	0.30	33.4	0.0	0.142	0.000
	0.40	33.5	0.0	0.133	0.000
	0.48	32.4	0.7	0.122	−0.015

**Table 6 materials-15-08656-t006:** Error analysis of the CLSCs predicted by RFCs–AFTCs.

Average Error for all Sample Strain Rates					
Temp.	Mean	Max	Temp.	Mean	Max
350	2.03	4.28	400	2.16	4.43
450	1.65	4.05	500	1.36	4.12
550	1.55	4.59			

## Data Availability

Not applicable.

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
