# Peer review of "Accurate Flow Characterization of A6082 for Precision Simulation of a Hot Metal Forming Process"

_materials, 2022, doi:10.3390/ma15238656_

Round 1

Reviewer 1 Report

Considering the friction and temperature effects, the flow behaviors of A6082 alloy are corrected, and the deformation characterization is discussed. The topic is interesting. Some minor errors need be reivsed before the possible accepted.

More recent references could be considered. The optimization of material parameters can refer to “Materials Science and Engineering: A 803, 140491”, "Materials Today Communication 32, 103855".

Author Response

  • Reviewer #1: Considering the friction and temperature effects, the flow behaviors of A6082 alloy are corrected, and the deformation characterization is discussed. The topic is interesting. Some minor errors need be revised before the possible accepted.

More recent references could be considered. The optimization of material parameters can refer to “Materials Science and Engineering: A 803, 140491”, "Materials Today Communication 32, 103855".

Ans. The following sentence with appropriate references were added:

“Recently, various artificial-neural-network models have been studied to characterize the hot deformation behaviors of metallic materials [27-30], which are deemed the promising methods of solving the hottest issue in this field.”

We appreciate the reviewer for kind advise.

Reviewer 2 Report

It would be a good thing to clarify the “Methodology” section a little bit. Please, specify the chemical composition of the alloy, name the model of the used equipment (country and manufacturer), indicate the number of the tested samples and the kind of the thermocouple that was used, etc.

Section 2 contains both the test procedure and the results of the experiment. It would be more convenient for the authors to separate these data into two separate sections. In the “Methodology” section, you should describe in more detail the methods of conducting experiments and describe a short work schedule. And in the third section, you may present the results of the work.

It is advisable to add the calculated data on the percentage discrepancy between the calculated results and experimental data to the “Conclusions” and “Discussion” sections.

Author Response

  1. It would be a good thing to clarify the “Methodology” section a little bit. Please, specify the chemical composition of the alloy, name the model of the used equipment (country and manufacturer), indicate the number of the tested samples and the kind of the thermocouple that was used, etc.

Ans. The following sentence were added:

“The material to be studied is an A6082 alloy (Chemical composition: Mn (0.4-1.0%), Fe (0.0-0.5%), Mg (0.6-1.2%), Si (0.7-1.3%), Cu (0.0-01%), Zn (0.0-0.2%), Al (Bal.)).”

“A specimen for each test case was tested since the test system has been adequately and strictly managed and well-controlled [31].”

“Solid cylinder compression tests were conducted using a Gleeble 3500 system. The diameter and height of the specimen are 10 and 15 mm, respectively. A specimen for each test case was tested since the test system has been adequately and strictly managed and well-controlled [31].”

  1. Section 2 contains both the test procedure and the results of the experiment. It would be more convenient for the authors to separate these data into two separate sections. In the “Methodology” section, you should describe in more detail the methods of conducting experiments and describe a short work schedule. And in the third section, you may present the results of the work.

Ans. We divided Section 2 into 2. Experiments, 3. Friction and temperature compensation.

  1. It is advisable to add the calculated data on the percentage discrepancy between the calculated results and experimental data to the “Conclusions” and “Discussion” sections.

Ans. The following sentence were added:

“However, it is interesting to note that the improvement of the RFCs-AFTC from RFCs-BFTC summarized in Table 3 can be neglected, implying that the increase in flow stress by the temperature-compensation (2.22% in the case of frictional coefficients of 0.35, strain of 0.46, strain rate of 1 s-1 and temperature of 350℃) is almost the same as its de-crease by the friction-compensation (2.49% in the same case with above).”

We appreciate the reviewer for kind advise.

Reviewer 3 Report

The research work presented by the authors is very complete, since they extensively studied experimentally and theoretically the effect of friction and temperature on the flow behavior of the A6082 alloy. In addition, they present a complete discussion of the different models used by other authors and relate them to a very clear qualitative discussion. In my opinion the article can be published as is.

Author Response

  • Reviewer #3: The research work presented by the authors is very complete, since they extensively studied experimentally and theoretically the effect of friction and temperature on the flow behavior of the A6082 alloy. In addition, they present a complete discussion of the different models used by other authors and relate them to a very clear qualitative discussion. In my opinion the article can be published as is.

Ans. We appreciate the reviewer for deep understanding of the authors’ work with encouraging words.

We appreciate the reviewer for kind advise.

Reviewer 4 Report

After studying the post, I give the following opinion. The post is well and logically processed. It is well illustrated and all the facts are explained in detail. I agree with the discussion, conclusions and other statements of the authors. I have no significant comments and, in my opinion, the contribution can be published in the submitted form.

The abstract is sufficient and describes exactly what the authors wanted and what they achieved. I highly value the introduction, where the entire spectrum of necessary facts and knowledge is analyzed in detail and exhaustively. In part 2, the authors describe in detail their research and the goals they want to achieve and what they are pursuing. I highly appreciate the nice photos of the samples in fig. 1. which are reliably illustrated by the experimental works of the authors. Graphs in fig. 2 illustrate the experimental process very well and it can be seen that the authors know the issue in detail. also table 1 testifies to their professionalism. This of course applies to all images and tables in the entire text of the post. The discussion is done at the required level and I, as a reviewer, have no comments on it. The conclusion is brief, but it is enough. 35 sources are listed in the references, which is sufficient for the presented area.

Author Response

  • Reviewer #4: After studying the post, I give the following opinion. The post is well and logically processed. It is well illustrated and all the facts are explained in detail. I agree with the discussion, conclusions and other statements of the authors. I have no significant comments and, in my opinion, the contribution can be published in the submitted form.

The abstract is sufficient and describes exactly what the authors wanted and what they achieved. I highly value the introduction, where the entire spectrum of necessary facts and knowledge is analyzed in detail and exhaustively.

In part 2, the authors describe in detail their research and the goals they want to achieve and what they are pursuing. I highly appreciate the nice photos of the samples in fig. 1. which are reliably illustrated by the experimental works of the authors. Graphs in fig. 2 illustrate the experimental process very well and it can be seen that the authors know the issue in detail. also table 1 testifies to their professionalism. This of course applies to all images and tables in the entire text of the post. The discussion is done at the required level and I, as a reviewer, have no comments on it. The conclusion is brief, but it is enough. 35 sources are listed in the references, which is sufficient for the presented area.

Ans. We appreciate the reviewer for deep understanding of the authors’ work with encouraging words.

We appreciate the reviewer for kind advise.

Reviewer 5 Report

Some comments were listed below for a further improvement of the paper.

(1) Why the friction-compensation deteriorates the accuracy of the modified flow curves ?

(2) How about the accuracy of the modified flow curves if only the temperature-compensation was applied? 

Author Response

  • Reviewer #5: Some comments were listed below for a further improvement of the paper.

  1. Why the friction-compensation deteriorates the accuracy of the modified flow curves?

Ans. According to our research, friction and temperature compensate for each other particularly in case of the aluminum alloy. If only one of them is compensated, the unbalance deteriorating the flow curve is inevitable.

  1. How about the accuracy of the modified flow curves if only the temperature-compensation was applied?

Ans. As explained above, it should deteriorate the fitted flow curves.

We appreciate the reviewer for kind advise.

Reviewer 6 Report

In the paper I found some inaccuracies that should be explained or corrected:

1. line 71: after ,,… C-m model” Authors should add literature [26]

2. Did the coefficient of friction influence the shape of samples?

3. What was a true strain?

4. Fig. 1. Authors should add the widths of the samples (min. and max.)

5. The max temperature was 550°C, while the melting temperature was 555°C. In Fig. 8 we can see that the increase of temperature as a result of plastic deformation is 16°C, where the strain rate was equal to 10 1/s. The question is about the increasing temperature for the variants at 550°C. There may probably appear a local melting area. That’s why Authors should add a metallographic analysis. Authors should add the additional Fig. with the temperature distribution at 550°C.

6. The next parameter for the model validation is the diameter of the samples after deformation. Authors should compare the FE and plastometric tests results.

7. Fig. 5. For higher strain rate values the error was increased. Authors should add some explanation

Author Response

  • Reviewer #6: In the paper I found some inaccuracies that should be explained or corrected:
  1. line 71: after ,,… C-m model” Authors should add literature [26].

Ans. We attached the reference and renumbered in order.

  1. Did the coefficient of friction influence the shape of samples?

Ans. Yes it does. However, the reduced contact interface (decreasing load) compensated for the barreled shape (increasing forming load).

  1. What was a true strain?

Ans. It is the logarithmic strain, accumulation of strain increment based on the current configuration.

  1. 1. Authors should add the widths of the samples (min. and max.)

Ans. We added the minimum and maximum diameters, and height at the final stroke.

  1. The max temperature was 550°C, while the melting temperature was 555°C. In Fig. 8 we can see that the increase of temperature as a result of plastic deformation is 16°C, where the strain rate was equal to 10 1/s. The question is about the increasing temperature for the variants at 550°C. There may probably appear a local melting area. That’s why Authors should add a metallographic analysis. Authors should add the additional Fig. with the temperature distribution at 550°C.

Ans. The following sentence were added:

“Note that the oscillation of compression load of the specimen with the initial temperature (550℃) near to the melting temperature could be clearly observed from Figure 2 as the strain rate and strain increased. The maximum measured temperature at the barreled surface reached 557℃, implying that internal local melting might occur during the compression test at the strain rate of 20 s-1.”

  1. The next parameter for the model validation is the diameter of the samples after deformation. Authors should compare the FE and plastometric tests results.

Ans. We added Fig. 9, comparing the predicted and experimental cylinder compression.

  1. 5. For higher strain rate values the error was increased. Authors should add some explanation.

Ans. The following sentence were added:

“Notably, the error between experimental flow curves and RFCs-AFC increases as the strain rate increase owing to relatively considerable oscillation of experimental flow curves caused by the increased temperature non-uniformity of the specimen.”

We appreciate the reviewer for kind advise.

Round 2

Reviewer 2 Report

My recommendations are taken into account by the authors. The article has been corrected.

Reviewer 6 Report

The paper is ready for publication.

I know the definition of true strain. I think my question was not clear. I asked about the value of the true strain in the plastometric tests. Not definition